# The impact of conducting preclinical systematic reviews on researchers and their research: A mixed method case study

Julia M. L. Menon[1,2]*, Merel Ritskes-Hoitinga[1,3], Pandora Pound[4], Erica van Oort[5]

**1** Department for Health Evidence, Systematic Review Centre for Laboratory (animal) Experimentation (SYRCLE), Radboud University Medical Center, Nijmegen, The Netherlands, **2** Preclinicaltrials.eu, Netherlands Heart Institute, Utrecht, The Netherlands, **3** Department for Clinical Medicine, AUGUST, Aarhus University, Aarhus, Denmark, **4** Safer Medicines, Kingsbridge, United Kingdom, **5** ZonMw (Netherlands Organisation for Health Research and Development), The Hague, The Netherlands

* Julia.menon@heart-institute.nl

**Data Availability Statement:** All relevant data are within the manuscript and its Supporting Information files.

## Abstract

### Background

Systematic reviews (SRs) are cornerstones of evidence-based medicine and have contributed significantly to breakthroughs since the 1980's. However, preclinical SRs remain relatively rare despite their many advantages. Since 2011 the Dutch health funding organisation (ZonMw) has run a grant scheme dedicated to promoting the training, coaching and conduct of preclinical SRs. Our study focuses on this funding scheme to investigate the relevance, effects and benefits of conducting preclinical SRs on researchers and their research.

### Methods

We recruited researchers who attended funded preclinical SR workshops and who conducted, are still conducting, or prematurely stopped a SR with funded coaching. We gathered data using online questionnaires followed by semi-structured interviews. Both aimed to explore the impact of conducting a SR on researchers' subsequent work, attitudes, and views about their research field. Data-analysis was performed using Excel and ATLAS.ti.

### Results

Conducting preclinical SRs had two distinct types of impact. First, the researchers acquired new skills and insights, leading to a change in mindset regarding the quality of animal research. This was mainly seen in the way participants planned, conducted and reported their subsequent animal studies, which were more transparent and of a higher quality than their previous work. Second, participants were eager to share their newly acquired knowledge within their laboratories and to advocate for change within their research teams and fields of interest. In particular, they emphasised the need for preclinical SRs and improved experimental design within preclinical research, promoting these through education and published opinion papers.

**Funding:** Funding for this study was provided by the Netherlands Organisation for Health Research and Development (ZonMw) (https://www.zonmw.nl/en/), reference number 50009817. It was awarded to MRH. ZonMw, represented here by Erica van Oort, played a role in study design, management and coordination of responsibilities in planning and reviewing of the manuscript.

**Competing interests:** I have read the journal's policy and the authors of this manuscript have the following competing interests: Julia Menon declares that she worked for the Department "Health Evidence" within the Radboudumc, in the same team as the coaches who provided training and support for the "knowledge infrastructure" module. Since February 2021, she has a paid position via ZonMw. Merel Ritskes-Hoitinga, who supervised this study, is the head of the SYstematic Review Center for Laboratory (animal) Experimentation (SyRCLE) team. Pandora Pound declared no competing interest. Erica van Oort, who supervised this study, is project manager for ZonMw and is in charge of the "knowledge infrastructure" module, as part of the ZonMw MKMD programme. However, we all sincerely declare that we did our utmost to remain impartial when conducting, analysing and supervising this study.

## Conclusion

Being trained and coached in the conduct of preclinical SRs appears to be a contributing factor to many beneficial changes which will impact the quality of preclinical research in the long-term. Our findings suggest that this ZonMw funding scheme is helpful in improving the quality and transparency of preclinical research. Similar funding schemes should be encouraged, preferably by a broader group of funders or financers, in the future.

## Introduction

Keeping up to date with health/medical the literature can be challenging due to the vast number of new articles published every year. Scholarly peer-reviewed journals produce over 3 million articles annually, with a 56% increase since the last decade [1,2]. Consequently, researchers, policymakers and healthcare providers require a way to systematically identify and evaluate literature on a specific topic. For 40 years, systematic reviews (SRs) have provided a powerful way of achieving this within clinical research. SRs follow clear and defined steps, and aim to provide a synthesis as well as a critical assessment of all the available relevant evidence [3]. Such syntheses enable researchers to identify research gaps and help future decision making in practice and policy. The methodology of SRs emerged from the evidence-based medicine paradigm of the 1980's [4,5]. Today, they are cornerstones of evidence-based medicine, with 30,000 SRs protocols being registered as of 2017 and a 2014 estimate putting the number of published SRs at over one million [6,7]. However, despite their advantages, SRs are struggling to achieve similar status in the preclinical field (i.e., fundamental and applied animal studies, in vitro and ex vivo studies before clinical research) [8]. A lack of knowledge, skills, or awareness of their value may be to blame. The first preclinical SRs began to appear in the early 2000s, a decade or so after the clinical SR standards were established [9]. Despite the slow start, preclinical SRs have been shown to increase transparency, avoid unnecessary duplication and help identify and improve poor reporting and poor study design [8,10,11]. They can be used retrospectively to cast light on clinical trial data or prospectively to prepare for new clinical and preclinical studies, and they may be used to guide future (translational) research [8,10,11].

Interest in preclinical SRs is slowly growing within academia and amongst other stakeholders [12–16]. Since 2011, the Netherlands Organisation for Health Research and Development (ZonMw) has invested in education and coaching in preclinical SRs through their funding programme "More Knowledge with Fewer Animals" (Meer Kennis met Minder Dieren in Dutch or MKMD) [16]. The main goal of this funding programme is to promote and implement the adoption of animal-free research methods by funding new animal-free (human based) innovations and encouraging the use of already existing alternatives. The programme consists of several modules, each with a specific focus [17]. The Knowledge infrastructure module focuses on providing preclinical SR education and coaching via workshops and training, and on promoting open access publishing of (negative or neutral) results [17]. Since its formation, 22 SR workshops have been organised within the Netherlands (over the period 2013–2020) and many participants have subsequently enrolled to conduct a coached preclinical SR.

While the benefits of clinical SRs are beyond doubt, the impact of conducting a preclinical SR on researchers and their research field remained unknown. Hence, to investigate these effects, we evaluated the impact of ZonMw funded preclinical SRs on researchers and (their) research.

### Research questions

Our study aims to assess the impact of conducting preclinical SRs on researchers, their research and their field through the following objectives:

1. What impact does conducting a preclinical SR have on a researcher in terms of planning, conducting, reporting and appraising their research projects?

2. What impact does conducting a preclinical SR have on a researcher's views about research, their own research, and their field more generally?

3. What impact does preclinical SRs have on preclinical research in general (e.g., quality, reproducibility, transparency, accessibility)?

## Theoretical framework

This project is a research impact case study aiming to evaluate the impact of an intervention (conducting preclinical SRs) on individuals (researchers), products (their research), and environment (their research field). Therefore, it relies on theories and knowledge of research impact assessment. An in-depth description of our theoretical framework can be found in the original ZonMw internal report made for this case study, available at: https://www.zonmw.nl/nl/actueel/nieuws/detail/item/systematisch-literatuuronderzoek-vervangt-vermindert-en-verfijnt-proefdieronderzoek/.

Briefly, research impact assessments rely on the evaluation of research impacts, defined as "contributions that research makes to the economy, society, culture, national security, public policy or services, health, the environment, or quality of life, beyond contributions to academia" [18]. These evaluations tend to be conducted to address one or several of the 4As: Advocacy, Accountability, Analysis and learning, and Allocation [19,20].

Overall, research impacts will lead to demonstrable and beneficial changes in behaviours, beliefs, and practices [21]. Consequently, these evaluations usually assess all directions and categories of impacts in a structured and high-quality manner [22]. This may include the use of complex, time-intensive frameworks, including impact pathways to identify or foresee how research projects create (expected) impacts, for example the societal or environmental impact of a research project [23,24]. Since impact evaluations began to be conducted by universities in the 1970's and 1980's, a plethora of frameworks has been proposed, with different scopes, aims, and assets [23–25]. Each proposes a specific approach to assessing research knowledge and research quality, and to measuring impacts [26]. Well-known frameworks include the Canadian Academy of Health Sciences Preferred Framework, the National Institute of Health Research Dashboard, or the Excellence in Research for Australia [24,27,28]. Such large endeavours were considered out of scope for our current project. Therefore, we chose two smaller frameworks to structure our study, namely 1) the research impact framework and 2) the behaviour wheel of change.

The research impact framework was developed in 2007 by Kuruvilla et al., as a checklist to guide researchers in selecting and evaluating the impacts of their work and interventions [29]. It highlights four areas of impacts: 1) research-related impacts, 2) policy impacts, 3) service impacts, and 4) societal impacts. We focused on the first area and designed our study using the seven categories it provides, which range from "providing data about a problem", "study replication", "innovation of new methods" to subtler impacts such as "becoming a member of a scientific society". All seven categories are available in detail in S1 Appendix.

The behaviour wheel of change created by Michie et al., in 2011 presents the idea that behavioural change occurs as a result of good pre-dispositions, interventions, and policies (which

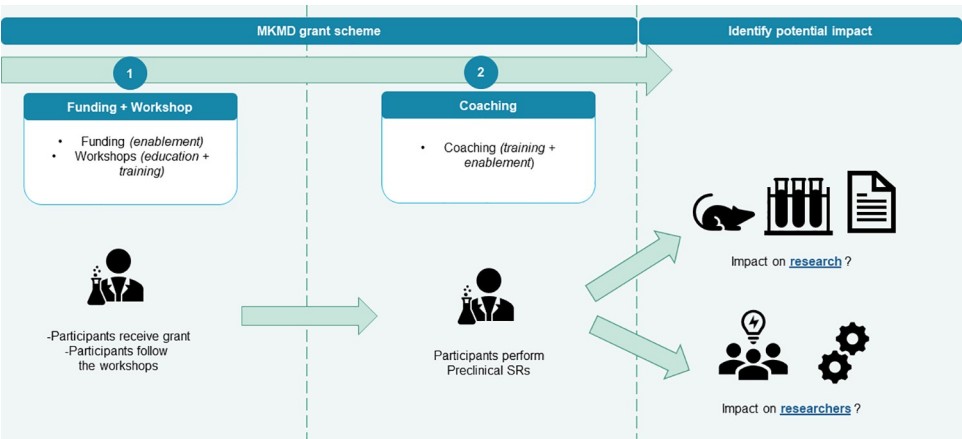

**Fig 1. Visual representation of the current hypothesis on impacts.**

enable or support the particular intervention and are created [in part] by responsible authorities] [30] (See S2 Appendix). These good pre-dispositions are capability, opportunity, and motivation as supported by the COM-B system—a behavioural change theory [30,31]. Within the intervention covered by this framework, three types corresponded to our case study: education, training, and enablement.

As a result, we hypothesise a pathway by which our intervention (conducting preclinical SRs) in a set context (supported by ZonMw's workshops, coaching and funds) would create impacts (for researchers and their research) (Fig 1).

## Materials and methods

This study complies with the standard for reporting qualitative research and the consolidated criteria for reporting qualitative research [32,33]. Checklists are available in S3 and S4 Appendices.

### Investigator characteristics and reflexivity

The primary investigator is a research assistant with a Master's Degree in Science, Innovation and Management (Radboud University). She has experience in preclinical SRs and qualitative research via her Master's Degree and two long-term Master's internships. She had no relationship with any of the participants and had not taken part in the training or coaching provided by the funding scheme being studied, although she had taken part in one of the workshops. Participants were not aware of her personal goals or personal motivation to conduct this research project.

Her interest in preclinical SRs and qualitative research led her to conduct this research project. She was supported and/or supervised by a research fellow with a background in ethics and philosophy, a professor in Evidence-Based Laboratory Animal Science and a ZonMw program manager, all of whom helped create a robust framework for the project.

### Context

This study uses a mixed-method approach combining questionnaires and semi-structured interviews. We focused on researchers who had followed ZonMw's workshops and who (had) received coaching to perform their own preclinical SR. This enabled us to get a snapshot of the

preclinical SR field and fairly evaluate the experience of these researchers within a given time period.

## Sampling strategy

We used purposive sampling to select our participants. The target population for the questionnaires comprised researchers who had participated in a ZonMw workshop and who had either started (currently conducting or prematurely stopped) or completed a preclinical SR with funded coaching. Those who had completed their SRs were approached to take part in an interview to evaluate the impact of the SR on their subsequent research.

## Recruiting participants

Participants were recruited by e-mail and received a reminder two weeks after the first invitation (both the invitation and reminder are available in S5 Appendix). All e-mail addresses were obtained from the information giving during coaching. Unfortunately, some email addresses were inactive, and participants could not be traced despite extensive efforts.

Due to the nature of this project, we did not aim to achieve data saturation but rather to collect as much data as possible within the given timeframe.

## Data collection methods

The questionnaires and semi-structured interviews were considered complementary, with the interviews providing more in-depth data about the impacts of preclinical SRs than the quantitative information generated by the questionnaires. The questionnaires provided quantitative information about the impacts of preclinical SRs, while the interviews provided more in-depth data. Data collection took place for a period of 6 weeks (23/07/2020–04/09/2020), with the online questionnaires being available for one month (23/07/2020–23/08/2020). Data analysis was performed for a period of almost one month (24/08/2020-18/09/2020).

## Online questionnaires

We designed and uploaded the questionnaire onto "Questionpro" (https://www.questionpro.com/), a free questionnaire platform. Our questionnaire consisted of both closed (dichotomous, multiple-choice or scaled) questions and open-ended questions. For rating questions, seven-point Likert scales were chosen (to avoid the bias created by using five-point Likert scales), with the extremities being "completely disagree" and "completely agree" [34]. The participants were divided into two groups; the "SR completed group" and the "SR started group". The groups were kept separate by skip logic. For the former group, five categories of impact were addressed 1) designing and planning experiments, 2) writing manuscripts, 3) appraising research, 4) skills gained as a result of conducting (steps of) the SR, and 5) experience with conducting (steps of) the SR (including, but not limited to, publishing experiences and wishing to perform further SRs (for the completed group)). For the latter group, only points 3, 4 and 5 were evaluated. At the end of the questionnaires, researchers in the "SR completed" group were also invited to participate in a semi-structured interview on the same topic. The full questionnaire is available in S6 Appendix.

## Semi-structured interviews

Researchers willing to participate in an interview were sent an informed consent form by e-mail and were given the opportunity to ask questions before signing (available in S7 Appendix). They were ensured that they could withdraw from the study at any point without any

consequences. The informed consent form was based on the World Health Organisation informed consent form template for qualitative studies (https://www.who.int/ethics/review-committee/informed_consent/en/).

The semi-structured interviews lasted about one hour and were conducted by teleconference using GoToMeeting. Only the participant and researcher were present on the call. Interviews were structured using a list of open questions on the participant's experience with their SR and the impacts they felt it had on their research, attitudes, and research field. (The interview guide can be found in S8 Appendix). Field notes were written during and/or after the interviews and were anonymised.

## Data processing & data analysis

Questionnaire data were exported from Questionpro to Excel. The "SR completed" and "SR ongoing" groups were analysed separately. Data analysis consisted of frequency counts for close-ended questions (with calculations of median and means), while open-ended questions were subject to content analysis. All analysis was conducted by one reviewer (JMLM).

The interviews were video recorded via GoTo Meeting, converted into mp3 files using VLC player, and subsequently transcribed verbatim using Express Scribe Transcription Software. All recordings were anonymised; random numbers were acquired from www.random.org. Transcripts were not returned to participants for comments and/or corrections because we did not want to bias their first answers (as this evaluation is performed for a funding agency). Thematic analysis of the transcripts was conducted by one reviewer (JMLM) in Atlas.ti (Version 8.4.15.0), with some themes emerging from the data and some deriving from the Research impact framework [29,35]. The coding tree and code organisation can be seen in S9 Appendix. Feedback options were not included for either participants' questionnaires or interviews findings.

## Ethical concerns pertaining to human subjects

According to Dutch law, research involving humans must be reviewed by the Central Committee on Research Involving Human Subject or a Medical Research Ethics Committee, if the study is subject to the Medical Research Involving Human Subjects Act [36]. Questionnaire research does not fall within this act and does not require ethical review, unless the questions are burdensome, intimate, or if completing the questionnaire is time-consuming [37,38]. In our case, participants were not patients, children, or vulnerable persons, and the topics addressed did not relate to their health, traumatic events or sensitive matters [39]. Furthermore, the time required to answer was short (maximum 15 minutes). The topics addressed in both questionnaires and interview guides posed no risks to the participants, and in particular no risk of physical or mental harm. For these reasons, we did not seek approval from an Institutional Review Board. In addition, we took several measures to ensure anonymity, confidentiality, and privacy, and obtained informed consent, complying with both qualitative research standards and the data protection act of 2018: all data were anonymised and assigned a random identification code; as noted above, an informed consent form was signed by interviewees; interview recordings were deleted 16 weeks after the interview, any mention of the participants' names, institutes, or any indicators that could threaten anonymity were omitted from the transcript, and; only the primary investigator (JMLM) had access to the unblinded data.

## Techniques to enhance trustworthiness

Thorough piloting was performed for both questionnaires and semi-structured interviews by seven researchers knowledgeable about SRs and from a variety of backgrounds and career

levels, namely research assistants (n = 2), PhD student (n = 1), and professors (n = 4). The main investigator and (most of) the co-authors jointly partook in the planning and design of the questionnaires and interview guide. In addition, we performed methods triangulation by using both questionnaires and interviews to answer the same questions, which increase the trustworthiness of our findings.

# Results

## Response rate

Of 99 potential participants, we were able to contact 95. Sixty-one participants started our questionnaire, and 45 completed it (i.e., 16 drop-outs), giving a response rate of 47.4% and a completion rate of 73.8% (definitions of response and completion rate can be found here [40]). An overview of participants per phase is available in Fig 2.

Of the 61 participants, 36 belonged to the "SR completed" group (i.e., they had published or submitted a manuscript of their preclinical SR). In comparison, 18 participants belonged to the "SR ongoing" group (i.e., still in the process of conducting their SR). Two participants did not complete their SRs due to time constraints, while the 5 remaining participants terminated the questionnaire before answering the question about the state of their review. Therefore the 61 participants were divided as such: SR completed (n = 36), SR ongoing (n = 18), SR stopped (n = 2), and NR (n = 5). Ten participants agreed to participate in interviews but only eight interviews were eventually conducted due to the unavailability of two researchers.

## Questionnaires

An overview of the questions and the number of respondents for each question can be found in S10 Appendix. The (anonymised) results of the questionnaire are available in S13 Appendix.

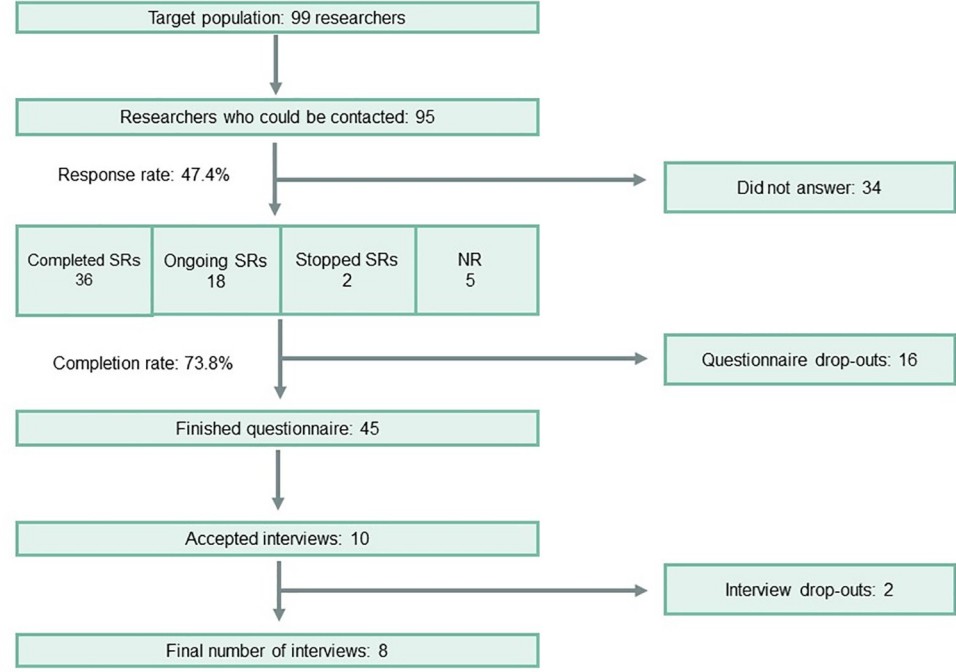

**Fig 2. Number of participants per phase.** Abbreviation: SRs: Systematic Reviews, NR: Not reported.

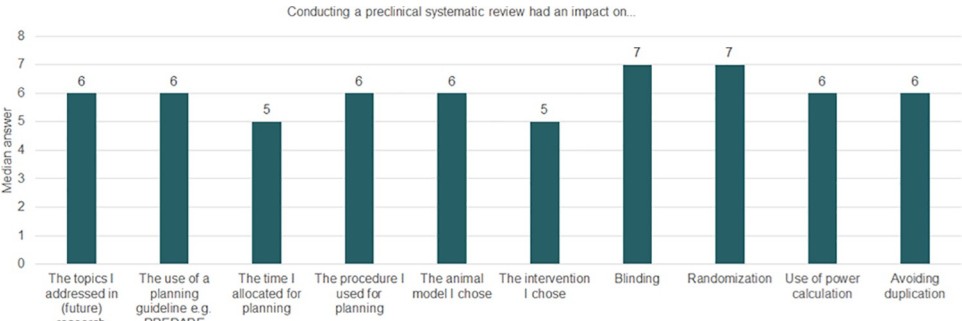

**Fig 3. Median answers on planning and designing subsequent research after conducting a preclinical SR.** This graph shows medians per question of a 7 points Likert scale. On this scale, 1 corresponded to completely disagree and 7 to completely agree. Thus, medians with values of 5 or above indicate agreement with the statements. Number of participants (n = 11).

**Impact on planning, designing, and writing research projects.** Within the SR completed group (n = 36), 14 participants went on to perform primary animal studies after completion of their SRs, 5 performed preclinical research using alternatives to animals, 6 performed clinical studies, and 6 moved to meta-research or ceased research. The remaining five participants did not complete this part of the questionnaire.

All participants who performed animal studies after their SRs answered questions about planning, designing, and reporting these studies. Most respondents agreed with the statements asked, as illustrated by the median of each question (Figs 3 and 4).

We found that conducting preclinical SRs impacted the way participants planned their future studies, in that they made more use of planning guidelines and increased the time allocated for planning and the use of power calculations to ensure statistical validity. They also contributed to avoiding unnecessary duplication. Moreover, conducting a preclinical SR also impacted the way participants performed their subsequent animal study, for example in terms of the animal model and intervention chosen, and use of blinding and randomisation. Similar results can be seen with regard to reporting. Conducting preclinical SRs strongly impacted the quality of reporting, including but not limited to increasing the use of reporting guidelines, the appropriate reporting of animal characteristics, housing conditions and methods (e.g., blinding and randomisation).

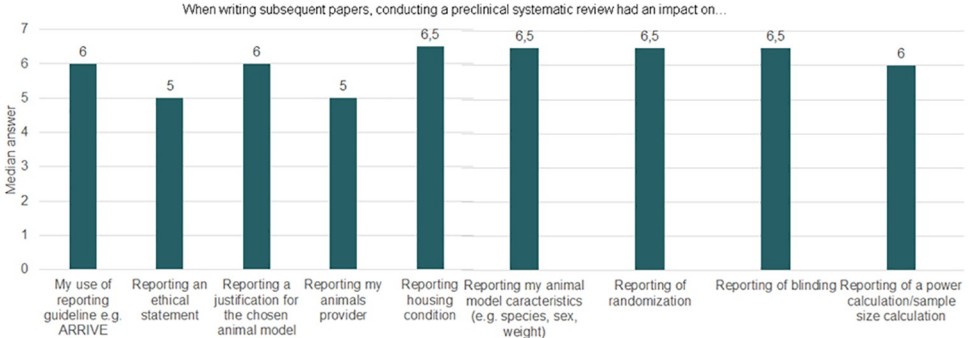

**Fig 4. Median answers on writing subsequent research after conducting a preclinical SR.** This graph shows medians per question of a 7 points Likert scale. On this scale, 1 corresponded to completely disagree and 7 to completely agree. Thus, median with values of 5 or above indicate agreement with the statements. Number of participants (n = 10).

Altogether, the results suggest that conducting a preclinical SR improved the quality of participants' subsequent animal studies through better and more thoughtful planning and conduct, as well as improved transparency as a result of better reporting of information and methods. (Detailed answers for each question can be seen in Fig 11.1 and 11.2 in S11 Appendix).

The SR ongoing group (18 participants) answered less detailed questions about their future research. However, all felt that conducting their preclinical SR would influence the way they would conduct and report their next experiment. Some suggested it would impact the model they would choose, the design of future *in vitro* studies, or would highlight data gaps for clinical research. Some participants felt there would be an impact on their field too. The following quotes illustrate these themes:

> *"Through the systematic review, we understand that the quality of currently published animal study reports is very poor, in particular, reporting randomisation and blinding. This reminds us to pay more attention to those aspects when designing, implementing and writing our own study."—Respondent 75156050*

> *"This study aims at investigating the best animal models that result in a similar outcome to human X. [. . .] Based on the findings of this study, I'd use the most robust models that will result in the reduction of the use of animals. I'm also confident that people who read this article will employ a similar approach and modify the design of their studies, thus overall, I think this study will have a big impact in the field"–Respondent 76823076*

Of the five people who performed research projects using non-animal methods, two affirmed that their SRs had positively influenced their choice of approach. The other three had already decided to use a non-animal approach prior to conducting their SRs. For the six who performed clinical research projects, three stated that the SR had impacted their clinical research; one with respect to the questions they chose to address, another in terms of translating preclinical to clinical research, and the third in terms of highlighting the need to investigate the role of a specific mutant phenotype

**Impact on research appraisal.** All participants answered questions about appraising research, regardless of the stage of their SRs. These questions aimed to determine the effect of conducting a preclinical SR on the way they conducted appraisals.

Overall, the results were equal across groups (Fig 5). Furthermore, few participants disagreed with the statements (data in Fig 11.3 in S11 Appendix). Our findings suggest that the critical stance of participants seemed sharpened by their experience with the preclinical SR, for both reading and assessing papers or research projects, and regardless of where that research comes from.

**Skills.** Participants from both groups stated that they gained new skills or improved existing skills in the process of conducting their preclinical SRs. Some of these skills are exclusive to research (e.g., meta-analysis skills, academic writing, critical appraisal), while others go beyond research (e.g., collaboration, negotiating with editors, interdisciplinary working). We identified and categorised four main types of skills: 1) research skills directly linked to the SR stages, 2) Research skills for planning/conducting subsequent research, 3) Critical appraisal, and 4) Interpersonal skills (S12 Appendix).

**Experience with their preclinical SRs.** Reviews (and reviews steps) often took longer than expected (26/30 participants for the completed group and 8/15 participants for the ongoing group). For instance, researchers often experienced the number of studies during screening, data extraction or data analysis as overwhelming, but also suffered in getting their

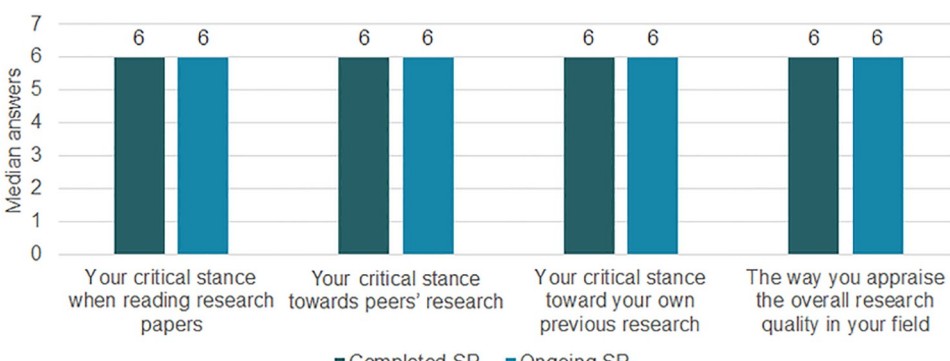

**Fig 5. Median answers on appraising research and critical stance after conducting (part of) a preclinical SR.** This graph shows medians per question of a 7 points Likert scale. On this scale, 1 corresponded to completely disagree and 7 to completely agree. Thus, medians with values of 5 or above indicate agreement with the statements. The total number of participants was 31 for the "Completed SR" group and 18 for the "Ongoing SR" group.

protocols, manuscripts or revision for publication reviewed. Some had to wait for second or third assessors/reviewers and experienced delays when asking authors for further information. Also, three participants (both groups combined) mentioned the COVID-19 pandemic as a delaying factor. Even though many participants experienced extensive delays while conducting their reviews, most considered it likely that they would conduct another SR or advise colleagues and peers to conduct a preclinical SR. Numbers wise, 34/46 participants mentioned that they would be likely to conduct another SR (18 "yes" and 16 "maybe"), while 45/46 would potentially recommend colleagues to do their own SRs (37 "yes" and 8 "maybe"). Eight participants from the completed SR group had already conducted a second SR after completing their first coached review, illustrating how useful they considered this. Additionally, in the SR completed group, 20/28 participants reported that they would like to receive similar coaching for their next review. Fifteen of these wanted a more personalised approach, with a focus on certain stages of the SR, while 5 wanted coaching for the whole process.

## Interviews

The interviews afforded greater insight into the sort of impact that conducting a preclinical SR has on both researchers and their research, as well as how this happens. We identified that impacts occurred in two steps and influenced three given levels (Fig 6).

First, impacts occur at the level of the researchers; conducting the SRs contributes to an evolution in participants' thinking, providing them with important insights and skills. This first step leads researchers to step back from their usual point of view and triggers the realisation of specific issues and shortcomings in their own work, research field, and science community.

Second, impacts on research can occur now that a new mind-set has developed. Researchers actively modify their activities 1) relative to their own work (*lab level*), 2) regarding how they appraise their field and advocate for change (*field level*), and 3) promoting change on a broader scale (*science community level*).

Both steps and levels of impact will be further explained and illustrated with empirical data in the upcoming sections.

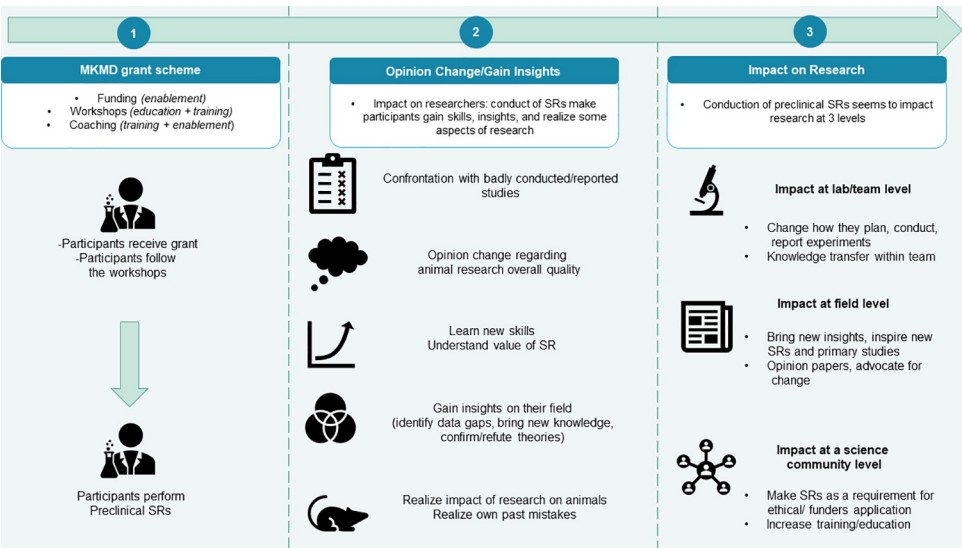

**Fig 6. Process and steps by which conducting preclinical SRs impacts researchers and research at different levels.**

**Step one. Impacts on researchers: A change in mind-set.** Interviewees were from various academic backgrounds and fields. However, they achieved similar insights, skills, and realisations as a result of conducting their preclinical SR.

Even though most respondents were already aware of the reproducibility crisis, conducting their review brought them face to face with poorly designed and/or poorly reported studies. This triggered an understanding, not only of the poor quality of animal studies in their field but about their own past mistakes:

> "I was quite shocked how poorly designed some studies are, while still being published in a high impact journal"–ZonMw 72

> "I had two medical students, and they helped me, also during the risk of bias, and they were like 'how researchers can be so stupid, they are so stupid', and I was like 'well, I think I did this too'."–ZonMw 23

The realisation occurred for most participants during the risk of bias assessment and data extraction stage, as many mentioned they were quite 'frustrated' due to unclear or missing information. This eye-opener on quality was confirmed when participants were unable to interpret studies properly or when these poor-quality publications impaired their final analysis.

> "We had a lot of data, the amount was more than enough but because the quality, I would say, of the studies is pretty low and sometimes you can't find the right information, it's harder to draw conclusions."–ZonMw57

As a result, participants reflected on their past flaws and learned how to prevent such mistakes in the future. For some, these issues made them rethink their activities in the preclinical field, their reasons for using animals and how they performed experiments:

*"There's a lot of bias that could be introduced in a model and having done it myself, I really realised that it's really pitiful of doing these experiments if you don't do them in the proper way."–ZonMw84*

*"It made me really more aware of why you [would] want to use animals and in what way. And even though in my own research I would want to do it in a good way, I saw that we also have flaws, and it made me more aware of what you're actually doing when you're doing animal research"- ZonMw57*

Despite insufficient quality or lack of evidence, most participants were able to draw conclusions and gain insights. We organised these insights under the four following categories: 1) increased awareness of their own field, 2) identification of data gaps and topics for new primary studies, 3) discovery of heterogeneity and incoherencies, and 4) discrediting or confirming theories.

1. *"You retrieve papers that you were not aware of even though you're quite familiar with literature on the topic, you will always find new things"–ZonMw80*

2. *"I really have more, much more transparent ideas about what studies are missing, and need to be done"–ZonMw23*

3. *"What was striking for this model is that everybody in the field was using the model in a different way."–ZonMw84*

4. *"We did encounter unexpected findings, which we couldn't even explain why it was the case. And it really contradicts some basic theories of (field of interest), of how our intervention works."–ZonMw 91*

Overall, participants were convinced about the value of preclinical SRs, and were positive about their experiences with the funding scheme including the coaching. Moreover, many participants admitted that they had underestimated the difficulty of the SR process and were glad to have received proper guidance.

*"I had underestimated really the efforts needed to make a strong and a good systematic review"–ZonMw72*

*"I really felt I needed this help"–ZonMw23*

**Step two. Impact on research (lab level).**   Research impacts were identified within the participants' teams and labs and were categorised as follows: 1) direct impacts on experiments, planning, conducting, reporting, appraising research; execution of other types of research, switching fields, conducting meta-research, and 2) advocacy for better research within their teams, promoting networking and collaboration.

The first category of impact was frequently mentioned in the questionnaires. Similar outcomes were reported in the interviews, namely that the planning, conduct and reporting of research projects was performed better after (or while) conducting the SR. The participants were unanimous about the change and progress they experienced thanks to their review:

*"I was just much more mindful about the blinding, randomisation, the sources of bias. We put enormous amount of efforts into doing that properly"–ZonMw91*

*"A lot of details would be missing if I hadn't done this (systematic review), cause now I know how much is missing in the studies"–ZonMw23*

Aside from doing new experimental research differently, participants' SRs fuelled ideas for other types of studies in meta-research and non-animal approaches. For some of them, the SR was a game-changer that led them to switch fields. Some now pursue their career in human research, some in meta-research (where they focus on SRs, meta-analysis, or the development of further tools) and some have distanced themselves from academia to work towards improving the system:

*"Step by step, you can do a lot of research in humans that is also very helpful. I did a lot of animal experiments and yeah (pause). I'm not going to do that anymore; I'm not going back to animal experiments"–ZonMw84*

*"I realised then that open science is what I'm interested in. That's also why I wanted to switch to a different working environment. Not doing research myself anymore that much but actually working on trying to improve the system"–ZonMw57*

The second category of impact emphasises our participants' enthusiasm about sharing their newly acquired knowledge with colleagues and peers. Some passed on their knowledge by teaching their new skills to students or by advising colleagues on how to improve their research, as the following quotes show:

*"I had this little extra time to teach students how to do it, and it becomes part of their package as well. Because in reality there's nothing about this type of science in education, and that's such a shame. Experimental design, statistics, and so, it's completely missed in education, and I feel that's what I want to teach my students, how to do good science"–ZonMw91*

*"I also used this to give presentations and make my colleagues at the department aware of how they should perform animal experiments. So, I try to implement this guideline of ARRIVE also in our department."–ZonMw84*

However, such improvements at a team or individual level may require changes from higher up in organisations. Several interviewees mentioned how supervisors' lack of familiarity with SRs could impede their execution and the transfer of skills and knowledge within a team. The fact that some professors and supervisors realised the value of SRs by the time they had completed the SR does, however, provide hope.

*"I have an assertive personality. [. . .] But many other people that are not as assertive might just not do a systematic review in the first place because of how many criticisms there is from the more managerial position in the department. Meaning professors themselves don't understand, then why should a PhD student do it?"–ZonMw91*

Resistance to change remains in the lab, both towards acceptance of preclinical SRs but also to other new methods that can improve research quality. Lab conventions, resistance from colleagues and supervisors, time pressure, funding and competition were the main obstacles experienced by the participants.

**Step two. Impact on research (field level).**   The two important ways by which our participants impacted their field with their preclinical SR were 1) by dissemination of their findings and 2) by promoting the value of SRs to improve the quality of research, for example, by writing an opinion paper.

The usual way of disseminating results is via publications and conferences. Interviewees' experiences highlighted that getting preclinical SRs published can be challenging. Some

participants had difficulties finding journals that would accept their SRs or, once found, faced repeated rebuttals. Moreover, the opinions of peer reviewers were quite influential in the editorial process. Responses about peer reviewers' input were mixed; some participants had a generally negative experience with peer reviewers who did not understand the SR process or the value of SRs, while others received beneficial input that improved their SRs:

> *"This is I think the biggest frustration of the whole process; that the journals and the reviewers really underestimate the significance, the work, that is in it.[. . .] they said no 'you have to do a letter to the editor, it's not worth an original article'–ZonMw23*

This dissonance directly echoes the lack of acceptance, support and education regarding preclinical SRs as mentioned earlier in "Step two. impact on research (lab level)". Nevertheless, once in the open, preclinical SRs can influence a field (see Box 1). Of course, publication does not ensure that peers will read or accept the SR, or implement the recommended changes. Some participants had frustrating experiences, discovering that the insights from their SRs appeared to have no impact on their field to date.

To press for change, certain participants chose an innovative route: some wrote opinion papers or letters to reach peers and editors, while some others took the matter a step further by creating tools (e.g., apps and websites) specific to their field.

> *"We wrote a paper in a XX journal for animal technicians, and describing all the differences that everybody does with the model. [. . .] also concerning housing and all kinds of things that*

---

**Box 1: Example of field impacts**

*EXAMPLE 1*

• A participant discovered that the mode of drug administration significantly impacted the results of an intervention, both in clinical and in preclinical experiments. With this issue identified, there could be a change in how this particular intervention is administered across the whole field.

*"With regard to the patient's studies, was that everybody just depleted these XX and saw what happened to XX, and then said, 'oh XX are bad' or 'oh XX are good'. But our review made I hope, make people realise when it gets published that it totally depends on **how** you administer your agent to deplete your XX. So, you can't really say 'oh they are good' or 'they are bad' because it depends totally on how you do it." – ZonMw23"*

*EXAMPLE 2*

• A participant's SR triggered a whole series of new SRs in the same direction as their own; as a result ofShowin the value and insight a preclinical SR can provide, other researchers in their field began performing their own preclinical SRs.

*"People really loved the paper and found it really informing to have a first good overview. And then, multiple research groups actually 'copy pasted' our paper, but put another cell type than the cell type we used [. . .] for the research field it was good, because everybody started to systematically assess all the evidences for certain cell lines"- ZonMw15*

---

*are important for animal studies. [. . .] I also wrote a paper for the XX organisation, about the use of animals and the 10 sorts of pitfalls, mistakes about experimental models"–ZonMw84*

**Step two. Impact on research (science community level).** Impacts at a broader level were also identified, i.e., increased awareness and acceptance of preclinical SRs, which could potentially improve the overall quality of preclinical research. These impacts included 1) the influence of stakeholders at a higher level and changing the status quo, 2) increasing training for researchers, and 3) improving and investing in the education of the next generation of researchers.

Observing that their SR findings did not bring about changes in their field, certain participants would have liked to pursue the issues at a higher level, for example by involving ethical committees and funders. It was suggested that preclinical SRs could become a requirement prior to performing an animal study and that this could be directed/checked by animal ethical committees. Funders could also be involved by emphasising the value of grant applications that include systematic overviews.

As noted above, questionnaire respondents were likely to recommend SRs to their colleagues and peers. This observation was confirmed in the interviews, with participants recommending that their peers or colleagues should receive proper training (via lectures, courses, and/or workshops) in the same way that they had. Participants made many suggestions about promoting preclinical SRs in their field, including advertising on websites, in universities, on social media and at conferences, and by providing financial support. It was suggested that more investment in education would be beneficial in terms of promoting the value and acceptability of SRs, since they do not appear to be sufficiently understood nor employed in many fields.

Finally, participants reflected on the younger generation of scientists, suggesting that preclinical SRs and related training would be beneficial for new researchers.

*"I think training is most powerful, because it's difficult to change the behaviour of the old researchers [. . .]. But the young researchers you can definitely train, they can become knowledgeable in this field."–ZonMw72*

## Discussion

Our findings have highlighted how conducting preclinical SRs impacted both research and researchers as a result of the ZonMw funding scheme, MKMD "Knowledge infrastructure" module. Both questionnaire and interview data indicated that conducting preclinical SRs provided researchers with awareness, skills, and insights that 1) were used and promoted in the planning, conducting and reporting of their subsequent projects, 2) influenced their critical stance when appraising research, and 3) were used and promoted in their research fields (e.g., teaching, dissemination of results, advocating for more transparency). Ultimately, these impacts could lead to more transparent, high-quality research. To our knowledge, this is the first time qualitative research has been employed to identify the research impacts of preclinical SRs.

At a lab level, we saw that conducting preclinical SRs not only provided insights and skills but also contributed to long-lasting behavioural changes amongst our participants. Furthermore, most participants were willing to improve their work and advocate for change in their teams (via support and teaching). We might extrapolate that if properly supported, coached, and empowered, a significant number of researchers might re-evaluate their research practices and follow the same path of reflection as our participants. This could lead to more teams wishing to improve knowledge transfer within their labs and among students, functioning a SR

feedback loop. Moreover, if universities invest in teaching about SRs and meta-analyses (e.g., on specific courses such as the legally required laboratory animal science courses for researchers wishing to perform animal studies (Function B course EU Directive 2010/63EU, Article 9 course in the Netherlands)), as suggested by our participants, this type of knowledge would become more universally accepted and might increase standards and expectations within the entire preclinical field.

However, knowledge transfer both within and between teams depends on several factors and antecedents, which can both promote and impede the dissemination of knowledge [41]. A legal framework and incentives from external stakeholders are needed to sustain this desire for improvement [30]. Given this context, incentives such as the MKMD funding scheme appear to provide appropriate support for initiating and sustaining improvements. Additionally, there are further external sources that could exert influence in this area. One of these is the ethical committee process. Both animal and human ethical committees could demand SRs prior to the conduct of new animal or human studies. Similar demands for transparency are happening with the pre-registration of preclinical studies, the value of which is increasingly acknowledged [42–44]. Other funding and regulatory bodies could further encourage the adoption of preclinical SRs by providing appropriate financial support or by promoting the value of preclinical SRs for translational purposes. Several organisations such as the UK's NC3Rs and NORECOPA already support the use of preclinical SRs [45,46]. Lastly, journals could promote the adoption of preclinical SRs if more editors were familiar with their value; to date, we found only four journals that regularly accept protocols for preclinical SRs [47–50].

Returning to the behaviour wheel of change framework, with both legislation and intervention in place, only the predispositions need to be set to trigger change. But as highlighted in the interviews, several factors currently hinder the adoption and acceptance of preclinical SRs, including standard lab conventions, resistance from colleagues and supervisors, time pressure, lack of funding, competition, and lack of awareness of their value. There are international hurdles too, such as journals not yet being willing to accept preclinical SRs and a lack of familiarity with SRs among journal editors and peer reviewers. Consequently, the emergence of preclinical SRs falls within a research culture paradox. On the one hand, high quality, transparent, and reproducible research is and should be standard. Previous research highlighted that (preclinical) SRs can play a useful role in this context as they can provide helpful information and insights for policy decision making and translation from preclinical to clinical studies [11,51,52]. On the other hand, researchers live in constant pursuit of delivering impactful results under great time pressure–leaving little space for conducting SRs, which are time-consuming [53]. However, and as this case study illustrates, the time invested is beneficial in that preclinical SRs seem to contribute to a change in mindset and behaviour and to improve research quality. Further studies are warranted to fully understand what hinders or facilitate the conduct of preclinical SRs. For instance, a more in-depth assessment of impacts would provide greater insight and could address other areas of the research impact framework, e.g., policy impacts, societal impacts. In this case study, we focussed primarily on our intervention, and it should be noted that the improvements observed cannot be only attributed to preclinical SRs as complex factors are also involved and may have influenced participants' answers (e.g., personal growth and capacity, background, seniority). Further assessments may help to highlight or understand the complex factors contributing to the beneficial effects.

Regardless of all positive aspects, we should bear in mind that SRs rely on the studies they include and may therefore be of limited value. Like primary studies, they may also be constrained by poor reporting or poor conduct. Such limitations are highlighted with respect to clinical systematic reviews, for example with protocols in PROSPERO not corresponding to the PRISMA-P guidelines, and outcome discrepancies between the protocol and final

publication [54,55]. Therefore, it is important that researchers continue to assess SR methodology and seek improvements [56–58]. On the preclinical side, Soliman et al., have provided new guidance for the appropriate conduct of SRs [59]. It is important to maintain a critical perspective and only use SRs where appropriate, and where they can be properly conducted and according to the guidelines.

## Strengths and limitations of the study

The major strength of this study lies in its mixed methodology, using both questionnaires and interviews to identify research impacts. This triangulation of data increases the validity of our findings. The choice of design and methodology was carefully considered by researchers from different academic disciplines, helping to ensure the study's robustness.

Regardless of a relatively limited set target population, we collected sufficient data using both questionnaires and interviews. The impacts we identified mapped onto the Research Impact Framework for the categories: "type of knowledge/problem", "research methods", "publication and papers", "translatability potential", "research network", and "communications", showing strong coherence between our findings and the Research Impact Framework. Our findings provide insight into how preclinical SRs have created impacts as a result of the ZonMw MKMD programme.

Despite significant strengths, we identified limitations in the methodology. First, our sample consisted mainly of Dutch researchers and was, of course, limited to researchers who were awarded a ZonMw grant within the knowledge infrastructure module. Consequently, our findings–enthusiasm and drive for improvements–may be related to the cultural context. The Netherlands are well known for their innovation-driven enterprise, so we cannot ensure that similar findings would emerge in a different context or with a foreign funding agency.

Second, our data collection took place during the COVID-19 pandemic and the study design we used limits the generalisation of our findings. Our findings are reported impacts, thus they are subjective to the perceptions of the participants, in contrast to measurable, tangible impacts. As mentioned earlier, the fact that (some of the) participants conducted their SR during the COVID-19 pandemic may have impacted their experience and influenced their responses. On the one hand, researchers might have had more time or opportunities to work on meta-research during the pandemic. On the other hand, the pandemic created delays in conducting the SRs and could have impacted the researchers personally, including their capacity and opportunity for professional development.

Moreover, our study design does not include a control group nor a pre-post intervention assessment, which limits the certainty one can have on the causality of our intervention. An example of this uncertainty can be observed in Fig 5, where both groups answered similarly regardless if they completed their review. This highlights again that the impacts and benefits found in our study cannot be only attributed to the conduct of preclinical SR but may be influenced by the workshop. We could imagine that similar education, i.e., workshops with hands-on practice, could produce similar results without actually conducting a preclinical SR.

Taken together, the intervention seems promising but our findings need to be interpreted within the narrow scope of the case study.

Lastly, because this study had to be conducted over the summer, it is possible that, despite our best efforts to obtain a high response rate, the summer holidays reduced the availability of participants. Furthermore, the time scale for this study was rather short (3 months), which limited the time we could spend collecting and analysing our data. Finally, the data collection, coding and analysis of the transcripts were all performed by one researcher. The participation of one or multiple additional investigators would have strengthened the trustworthiness of our

findings and potentially provided other interpretations. Investigator triangulation should be advocated when possible.

Nevertheless, to our knowledge, this is the first case study on such a topic, and in the future, when preclinical SRs become more established, it would be interesting to repeat this study on a broader scale, for example within European institutes and/or the Ensuring Value in Research Funder Forum. In addition, a pre-post test study could be performed to evaluate change in skills and behaviour before and after the intervention. A wider population would also allow analysis by demographic group, e.g., the comparison of senior vs younger researchers, comparison between research field.

## Conclusion

This case study has provided important insights into the impacts of training, coaching and conducting preclinical SRs on both research and researchers. Not only did SRs impact research at a lab level, they led to changes in researchers' views and critical abilities, and spurred efforts to advocate for improvement in their fields. Our project highlights the necessity and importance of supporting preclinical researchers to perform SRs, and demonstrates the impact of this on the quality and transparency of research, as well as on researchers' awareness and motivation to change the status quo. Our findings suggest that support such as that provided here by the Dutch funding agency ZonMw, is relevant and should be encouraged on an international scale, to improve the quality and translation of preclinical research.

## Supporting information

**S1 Appendix. The Research Impact Framework.**
(PDF)

**S2 Appendix. The Behaviour Change Wheel.**
(PDF)

**S3 Appendix. Standards for Reporting Qualitative Research (SQRQ) checklist.**
(PDF)

**S4 Appendix. Consolidated criteria for reporting qualitative studies (COREQ): 32-item checklist.**
(PDF)

**S5 Appendix. Invitation to complete the online questionnaire and reminder.**
(PDF)

**S6 Appendix. Full questionnaire.**
(PDF)

**S7 Appendix. Informed consent form.**
(PDF)

**S8 Appendix. Interview guide.**
(DOCX)

**S9 Appendix. Code tree and harmonisation.**
(PDF)

**S10 Appendix. Organisation of the questionnaire and number of respondents per questions.**
(PDF)

**S11 Appendix. Questionnaire results for questions on planning, designing and reporting animal experiments, and appraising research after conducting a preclinical SR.**
(PDF)

**S12 Appendix. Skills learnt and improved by performing preclinical systematic review.**
(PDF)

**S13 Appendix. Questionnaire results.**
(XLSX)

## Acknowledgments

We would like to thank Janne Swinkels, Stevie van der Mierden, Franck Meijboom, Brett Lidbury and Milan Khanpour, for piloting and providing feedback on questionnaire and/or interview guide. As well, we thank Rob de Vries for his support in triangulation and his feedback on the questionnaire and interview guide.

## Author Contributions

**Conceptualization:** Merel Ritskes-Hoitinga, Erica van Oort.

**Data curation:** Julia M. L. Menon.

**Formal analysis:** Julia M. L. Menon, Pandora Pound.

**Funding acquisition:** Merel Ritskes-Hoitinga.

**Investigation:** Julia M. L. Menon.

**Methodology:** Julia M. L. Menon, Merel Ritskes-Hoitinga, Erica van Oort.

**Project administration:** Erica van Oort.

**Supervision:** Merel Ritskes-Hoitinga.

**Visualization:** Julia M. L. Menon.

**Writing – original draft:** Julia M. L. Menon.

**Writing – review & editing:** Julia M. L. Menon, Merel Ritskes-Hoitinga, Pandora Pound, Erica van Oort.

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
