## [Decision Letter · Decision Letter 0]

2 Aug 2021

PONE-D-21-16684

The impact on research and researchers of conducting preclinical systematic reviews: a mixed methods case study

PLOS ONE

Dear Dr. Julia Menon,

Thank you for submitting your manuscript to PLOS ONE. This is generally a well written paper but after careful consideration, we feel that it has merit but does not fully meet PLOS ONE’s publication criteria as it currently stands. Therefore, we invite you to submit a revised version of the manuscript that addresses the points raised during the review process.

Please address the peer reviewers' comments which offer suggestions for improving readability and grammatical checks. Where applicable, we ask you to provide a point by point response the comments provided. As PLOS ONE does not copy edit accepted manuscripts we ask you to conduct a thorough spell and grammatical check before resubmission

We look forward to receiving your revised manuscript.

Kind regards,

Eleanor Ochodo

Academic Editor

PLOS ONE

2. Thank you for stating the following in the Competing Interests/Financial Disclosure * (delete as necessary) section:

“I have read the journal's policy and the authors of this manuscript have the following competing interests:

Julia Menon declares that she worked for the Department “Health Evidence” within the Radboudumc, in the same team as the coaches who provided training and support for the “knowledge infrastructure” module. Since February 2021, she has a paid position via ZonMw.

Merel Ritskes-Hoitinga, who supervised this study, is the head of the SYstematic Review Center for Laboratory (animal) Experimentation (SyRCLE) team.

Pandora Pound declared no competing interest.

Erica van Oort, who supervised this study, is project manager for ZonMw and is in charge of the “knowledge infrastructure” module, as part of the ZonMw MKMD programme.

However, we all sincerely declare that we did our utmost to remain impartial when conducting, analysing and supervising this study.” 

We note that you received funding from a commercial source: [Name of Company]

4. Please include a copy of Table 1 which you refer to in your text on page 33 and table 12 in page 34.

**Reviewers' comments:**

Reviewer's Responses to Questions

**Comments to the Author**

1. Is the manuscript technically sound, and do the data support the conclusions?

Reviewer #1: Yes

Reviewer #2: Yes

2. Has the statistical analysis been performed appropriately and rigorously? 

Reviewer #1: Yes

Reviewer #2: N/A

3. Have the authors made all data underlying the findings in their manuscript fully available?

Reviewer #1: Yes

Reviewer #2: Yes

4. Is the manuscript presented in an intelligible fashion and written in standard English?

Reviewer #1: Yes

Reviewer #2: No

5. Review Comments to the Author

Reviewer #1: 1. Yes, this manuscript technically sound, and its data support the conclusions reached.

2. Yes, statistical analysis were performed appropriately

3. Yes, all data was available

4. Yes, it is presented in an intelligible fashion and written in standard English

5. Comments have been attached.

Reviewer #2: I would like to thank the editor for the opportunity to review this very interesting paper; and would also like to thank the authors for conducting and reporting on this important and fascinating research topic on preclinical evidence synthesis; that is, relating to meta-research of laboratory experiments on animals. The paper investigates the impact of conducting a preclinical systematic reviews (SR), with funded coaching and workshops, on researchers and their subsequent research. While the limitations in terms of generalisability should be, and are, acknowledged, this is an important first step in reducing animal waste while improving and standardising preclinical work.

The work shines the light on a very important and often neglected aspect of research waste: the waste of animal subjects in research. The authors are commended for this. It is our responsibility as scientists to conduct research on animals as ethically, rigorously and efficiently as possible; to ensure we obtain the correct results the first time, with minimal animal use. This may very well be a first-order seminal paper in the establishment of SRs as the top tier of the evidence hierarchy in this field. I offer the following thoughts with regards to the manuscript for the consideration of the editor and authors. I have marked the most pressing issues, ones I would classify as major in a differently designed study, with 'IMPORTANT' throughout the review.

Overall

The manuscript would benefit from editing for minor spelling, punctuation and grammatical errors throughout, but readability in general is good. I have attempted to identify the most glaring errors in this review.

Abstract

Lines 17-18: It is not clear from this statement whether the focus of the ZonMw grant scheme is preclinical SRs specifically, or SRs in general.

Lines 22-23: This sentence does not accurately reflect that recruited researchers could be those who had finished a preclinical SR as well as those still busy conducting a preclinical SR (and indeed, those who had started, but not finished their SR).

Lines 36-39: Though the authors do couch their language here, a word of caution: in the absence of measurements in a control group which did not receive the funded interventions and conducted a preclinical SR; the authors are requested to exercise caution in attributing the reported changes to the interventions alone. Even though researchers may themselves attribute these changes to the intervention, there may be a complex interplay of factors, e.g. the growth that happens during the work done for a PhD, contributing to their changing views and skill sets.

Introduction

General: It would be useful if authors could explicitly define early on in the manuscript what is meant by ‘preclinical’, as the term is also sometimes used to refer to pre-hospital and emergency medicine contexts.

Line 41: Suggest changing to ‘Keeping up to date with health/medical literature…’

Line 46: Please remove errant ‘-‘ from the sentence ending on this line.

Line 56: Consider changing ‘throw light’ to ‘cast light’.

Line 74: This introductory line should be presented as such. In its present state it is confusing and appears as another question, instead of the overarching aim. I would suggest amending this sentence to ‘This work aims to assess the impact of conducting preclinical SRs on researchers, their research and their field through the following objectives:’

Lines 115-116: While it is acknowledged that these would be outside the scope of the current manuscript, which already describes a great deal of work, it would be interesting to assess the policy impacts and societal impacts (areas 2 and 4 in the manuscript) of the intervention in question using the modified Kuruvilla et al. 2007 checklist.

Materials and Methods

Line 142: It is not clear what is meant by ‘thrive’ in this context, but it is assumed to be an incorrect translation. Consider changing to ‘drive’ or ‘development’, depending on the original intent.

Lines 157-158: This sentence contradicts methodology in following sections, which describes participants who had not finished their SRs also included in the sample. Please correct this factual inaccuracy, or elaborate on why non-completers were included post hoc – if that was the case.

Line 163: It may be useful to include here whether participants consented to be contacted by ZonMw for the purposes of intervention assessment when providing details, if this was indeed the case.

Line 177: It would be useful, though not essential, to state at the start of this section whether informed consent was sought to use the responses provided in questionnaires.

Line 186: As it is not possible for researchers still conducting their SRs to evaluate post-SR skills and experience, it is assumed that this should read ‘For the latter, only points 1, 2 and 3 were evaluated.’ Please correct if this is the case, or elaborate on how post-intervention experiences would be appraised before the end of the intervention – if the manuscript is correct in its current format.

IMPORTANT Lines 205-208: It is a pity that more demographic/explanatory variables were not collected on participants, as some measures of association could have been calculated. This would have provided some insights into differences between researchers and how this may, or may not, have shaped their responses – specifically given the outstanding uncertainty presented by the lack of a control group of questionnaire participants. It would be useful if the authors could include these considerations in their manuscript, perhaps as a next step in the evolution of understanding the role of SRs in preclinical work.

Lines 235-236: Please provide a breakdown of the number of inputs per background/career level, i.e. ‘…Thorough piloting was performed for both questionnaires and semi-structured interviews by seven researchers knowledgeable in SRs with varying backgrounds and career levels, namely research assistant (n=x), PhD student (n=y) and professor (n=z).’ I don’t think it’s necessary to identify a person at a professorial level as ‘associate’ (or not).

Results

Lines 243-244: It is very possible that response and completion rates may be defined differently by different sources. A leading survey software defines response rate as ‘the number of people who completed the survey divided by the total sample’ (i.e. 45/95=47.4%), and completion rate as ‘the number of surveys filled out and submitted divided by the number of surveys started’ (i.e. 45/61=73.8%). Please revisit these concepts and calculations, and provide a reference for the definitions used to avoid confusion among readers.

Line 248: Please revise the start of the sentence ‘And the 7 remaining…’. It is recommended that this sentence be combined with the previous sentence.

Line 265-266: I am not convinced that reporting an ‘average median’ is wise; it is also a meaningless value in terms of agreement without information on the statements asked and the direction of the question. I wouldn’t risk readers skimming over this without referring to the figures, where the median breakdown per question provides much richer information.

Line 317: Steer clear of statements such as ‘the results are quite positive’ as this indicates inherent bias for a result.

IMPORTANT Lines 317-321: As stated before, it is very difficult to know whether the equivalence in the response of SR completers and non-completers can be ascribed to the SR, given the lack of a control comparator. It is possible that these insights may have evolved in respondents in a similar field, and with comparable education and training, in the absence of conducting a preclinical SR. This evolution might even be suggested by the identical medians in the two groups, with further SR experience (in completers) leading to no increase in critical appraisal compared to non-completers. It is accepted that this uncertainty is part of the case study design, but please make this clear to the reader.

IMPORTANT Line 338: It would be interesting to know whether respondents participating in the scheme during 2020 had a different experience to those pre-pandemic in terms of time taken/delays with their SRs. If these data are available this might be an interesting aspect to include, as the time required to conduct an SR often presents a considerable barrier to their initiation and completion.

Lines 470 and 484: Please steer clear from informal language such as ‘no easy business’ and ‘budge one bit’. Furthermore, ‘(yet)’ [line 484] again demonstrates an inherent bias by the author for a certain outcome, and should be avoided at all costs.

Discussion

IMPORTANT General: As stated before, it is very difficult to know whether the reported experiences and skills of respondents can be ascribed to conducting the preclinical SR, given the lack of a control comparator or association with explanatory variables. It is possible that the reported insights would have developed in respondents in a similar field and with comparable education and training. The authors do address the subjective nature of their findings, but could suggest next steps in this important work. As a starting point, it might be useful to suggest (and consider) a pre-post test with a new intake of researchers, measuring their attitudes and skills at baseline and again following the completion of their SR. An alternative, but a weaker test of causality, would be to measure explanatory variables and test associations with reported outcomes in future work.

General: It was surprising not to see any exploration of the effect of the pandemic on the findings presented in this paper, given both the global impact of the event as well as several mentions of COVID-19 in the questionnaire responses – specifically related to delays in SR completion. It might be useful for the authors to include a paragraph on how this context may have shaped their findings, both in terms of practicalities for SRs as well as responses; particularly due to shifts in mental health and outlook for many during this time.

IMPORTANT Lines 568-569: Statements attributing changes in mindset and behaviour to the conduct of preclinical SRs without acknowledging the complex myriad of circumstances that may contribute to these are problematic. Please review this statement (and others like it) to accurately reflect that preclinical SRs *appear to contribute* to a shift in mindset and behaviour, though comparative data, and more work to identify other explanatory factors, is needed.

Conflicts of interest

The authors declare their association with the invention provider upfront, and no serious conflicts were identified. I would like to caution the authors, however, around the language they use and how it unintentionally reflects a preferred direction of their findings.

With thanks again for the opportunity, and best wishes.

6. PLOS authors have the option to publish the peer review history of their article (what does this mean?). If published, this will include your full peer review and any attached files.

Reviewer #1: No

Reviewer #2: **Yes: **Dr Amanda S Brand

---

## [Author Response · Author response to Decision Letter 0]

10 Sep 2021

Dear Editors,

We would like to thank the editor and the two reviewers for their comments and suggestions and the opportunity to revise our submitted manuscript. 

We have addressed all issues that were raised and address detailed answers below. The comments were very insightful and have helped to improve our manuscript. 

We genuinely hope that our revisions will be satisfactory and that our manuscript will be now suitable for publication. 

Thank you again for considering our manuscript for publication in PLoS One. 

Looking forward to hearing from you again,

Sincerely,

On behalf of all co-authors,

Julia Menon

Editor’s comments:

1. Please ensure that your manuscript meets PLOS ONE's style requirements, including those for file naming

Authors’ response: we made a thorough check and made the necessary changes to comply with PLOS ONE’s style requirements. We genuinely apologise that several mistakes passed through and thank the editor for their keen eye and recommendation to modify our manuscript accordingly. Here is a detailed list of what was modified: 

-Page 1: Affiliations are numbered, the postal address of the corresponding author was removed, and we added initials to the corresponding author. 

-line 130: we change the title of Fig 1 to bold

-All headings and subheadings were verified and now comply with sentence case

-Appendices captions were remodelled for appendix S2, S11, and S12. The appendices files were all renamed as “SX_Appendix.pdf” for example “S1_Appendix.pdf”. 

-Appendices are now formatted as level 1 headings and their caption were verified for compliance. Appendices were changed from .docx to .pdf format. Therefore, we also aim to reupload all appendices to be consistent.

2. We note that you received funding from a commercial source: [Name of Company]

Authors’ response: We think a slight confusion may have occurred as ZonMw is not a commercial source. It is a Public Benefit Organisation, which provides and receives only public money.

To clarify, we still modified the competing interest statement. Therefore, please find below the new Competing Interest Statement:

“I have read the journal's policy and the authors of this manuscript have the following competing interests:

Julia Menon declares that she worked for the Department “Health Evidence” within the Radboudumc, in the same team as the coaches who provided training and support for the “knowledge infrastructure” module. Since February 2021, she has a paid position via ZonMw.

Merel Ritskes-Hoitinga, who supervised this study, is the head of the SYstematic Review Center for Laboratory (animal) Experimentation (SyRCLE) team.

Pandora Pound declared no competing interest.

Erica van Oort, who supervised this study, is program manager for ZonMw and is in charge of the “knowledge infrastructure” module, as part of the ZonMw MKMD programme.

ZonMw funded this project, as declared in our financial statement, as well as the employment of Erica van Oort. ZonMw is a Public Benefit Organisation, which is registered with the Chamber of Commerce The Hague under number 27365263, tax number: 0028.76.528. This does not alter our adherence to PLOS ONE policies on sharing data and materials.

However, we all sincerely declare that we did our utmost to remain impartial when conducting, analysing and supervising this study.”

Authors’ response: It seems our ethics statement only appears in the Methods section (in the subheading “Ethical concerns pertaining to human subjects”). 

4. Please include a copy of Table 1 which you refer to in your text on page 33, and Table 12 on page 34

Authors’ response: Table 1 and Table 12 are provided in Appendix S1 and S12. Consequently, we deleted their caption and only kept the captions of Appendix S1 and S12. For consistency reasons, we would rather have all supplementary materials be referred to as “Appendix”, than have a mix of tables, figures, supplementary materials etc, in the captions. We also modified the mention of Table 12 to “S12 Appendix” in line 335.

Reviewer #1 comments: 

1. Line 1-2: Please revise it: "The impact of conducting preclinical systematic reviews on researchers and their research: a mixed method case study" will sound good. 

Authors’ response: We agree that this title is more straightforward. We modified it accordingly. 

2. Line 15: Delete "a"

Authors’ response: the “a” was removed accordingly, and a “s” was added to “cornerstone”. Thank you for noticing this typo. 

3. Line 46: Delete "-"

Authors’ response: It was deleted accordingly. 

4. Line 49: Remove "a"

Authors’ response: the “a” was removed accordingly and a “s” was added to “cornerstone”. Thank you for noticing this typo. 

5. Line 50: Include also the number of SR being conducted.

Authors’ response: Thank you for this great suggestion. We added numbers of SR being conducted as of 2014 (this is the most recent update we could find on that matter) and added the reference to the relevant article. The text now reads: “Today, they are cornerstones of evidence-based medicine, with 30,000 SRs protocols being registered as of 2017 and a 2014 estimate putting the number of published SRs at over one million [6,7]”. 

6. Line 70: Remove "preclinical"

Authors’ response: we removed “preclinical” as you suggested. 

7. Line 161: Replace "contacted" with "recruited"

Authors’ response: we changed the sentence as suggested. 

8. Line 222-224: Even though this study did not pose any risk to participants, ethical clearance was needed since it involved human beings. This study should have applied for ethical clearence.

Authors’ response: we realise that our statement may have been unclear. So, we provide more information here and in the text of the manuscript. 

According to Dutch law, studies including humans can be subjected to the Medical Research Involving Human Subjects Act (WMO) and must be reviewed by the Central Committee on Research Involving Human Subject (CCMO) or a Medical Research Ethics Committees (MREC). However, some studies in which humans are participating are not subject to the WMO, and hence do not require ethical review (see here for more information: https://english.ccmo.nl/investigators/legal-framework-for-medical-scientific-research/your-research-is-it-subject-to-the-wmo-or-not and https://english.ccmo.nl/investigators/additional-requirements-for-certain-types-of-research/non-wmo-research ). 

Questionnaire research does not follow the WMO and hence does not require ethical review. The exception is if questions are, for instance, burdensome, intimate, or if the questionnaire takes a lot of time to fill in (https://english.ccmo.nl/investigators/additional-requirements-for-certain-types-of-research/other-types-of-research/questionnaire-research). Considering the nature of our questions and the short amount of time needed to fill the questionnaire, our research does not fall within that description. 

For clarity purposes, we added the following reference and modifications to the text: 

“According to Dutch law, research involving humans must be reviewed by the Central Committee on Research Involving Human Subject or a Medical Research Ethics Committee, if the study is subject to the Medical Research Involving Human Subjects Act [36]. Questionnaire research does not fall within this act and does not require ethical review, unless the questions are burdensome, intimate, or if completing the questionnaire is time-consuming [37, 38]. In our case, participants were not patients, children, or vulnerable persons, and the topics addressed did not relate to their health, traumatic events or sensitive matters [39]. Furthermore, the time required to answer was short (maximum 15 minutes). The topics addressed in both questionnaires and interview guides posed no risks to the participants, and in particular no risk of physical or mental harm. For these reasons, we did not seek approval from an Institutional Review Board.”

9. Line 242-248: Please add demographic characteristics of the participants; gender, age and their research levels (if possible).

Authors’ response: We did not collect demographic characteristics systematically. Therefore, we cannot add them to the text. However, we agree that demographic characteristics could have influenced the results and discuss their potential influence in our discussion. We also suggest that future studies use demographic groups (e.g. young researchers vs senior researchers)

Here is the text we added to our discussion:

 “In this case study, we focussed primarily on our intervention, and it should be noted that the improvements observed cannot be only attributed to preclinical SRs, as complex factors are also involved and may have influenced participants’ answers (e.g., personal growth and capacity, background, seniority). Further assessments may help to highlight or understand the complex factors contributing to the beneficial effects.” 

And also:

 “A wider population would also allow analysis by demographic group, e.g., the comparison of senior vs younger researchers, comparison between research field.”

10. Line 242-248: Please rephrase the paragraph it is confusing (not clear) in the current form, in terms of the numbers.

Authors’ response: We reverified all numbers and identified that an error got introduced, which may be the reason for the unclearness. It was not 7 but 5 participants who terminated the questionnaire. It was modified appropriately.

Moreover, we provided an additional sentence to clarify the numbers in the text and hope it will be suited to your expectations. At the end of the explanation, we state: “Therefore the 61 participants were divided as such: SR completed (n=36), SR ongoing (n=18), SR stopped (n=2), and NR (n=5).” 

The whole paragraph was also modified for more clarity:

Of 99 potential participants, we were able to contact 95. Sixty-one participants started our questionnaire, and 45 completed it (i.e., 16 drop-outs), giving a response rate of 47.4% and a completion rate of 73.8% (definitions of response and completion rate can be found here [40]). An overview of participants per phase is available in Fig 2.

Of the 61 participants, 36 belonged to the “SR completed” group (i.e., they had published or submitted a manuscript of their preclinical SR). In comparison, 18 participants belonged to the “SR ongoing” group (i.e., still in the process of conducting their SR). Two participants did not complete their SRs due to time constraints, while the 5 remaining participants terminated the questionnaire before answering the question about the state of their review. Therefore the 61 participants were divided as such: SR completed (n=36), SR ongoing (n=18), SR stopped (n=2), and NR (n=5). Ten participants agreed to participate in interviews but only eight interviews were eventually conducted due to the unavailability of two researchers.”

11. Line 260-262: What happened to the 11 participants in the SR completed group? 36 - 5 - 6 = 25. Please talk about the other missing 11 participants.

Authors’ response: We precise the number by adding the following sentences: “Within the SR completed group (n=36), 14 participants went on to perform primary animal studies after completion of their SRs, 5 performed preclinical research using alternatives to animals, 6 performed clinical studies, and 6 moved to meta-research or ceased research. The remaining five participants did not complete this part of the questionnaire.”

Reviewer #2 comments: 

1. Lines 17-18: It is not clear from this statement whether the focus of the ZonMw grant scheme is preclinical SRs specifically, or SRs in general.

Authors’ response: Indeed, the sentence could be confusing. Thank you for spotting it. We added “preclinical” in front of “SRs” to specify that the ZonMw grant scheme focuses on preclinical SRs specifically. The sentence now reads: “Since 2011 the Dutch health funding organisation (ZonMw) has run a grant scheme dedicated to promoting the training, coaching and conduct of preclinical SRs.”

2. Lines 22-23: This sentence does not accurately reflect that recruited researchers could be those who had finished a preclinical SR as well as those still busy conducting a preclinical SR (and indeed, those who had started, but not finished their SR).

Authors’ response: To bring more clarity, we rephrased as follow: “We recruited researchers who attended funded preclinical SR workshops and who conducted, are still conducting, or prematurely stopped a SR with funded coaching.”s

3. Lines 36-39: Though the authors do couch their language here, a word of caution: in the absence of measurements in a control group which did not receive the funded interventions and conducted a preclinical SR; the authors are requested to exercise caution in attributing the reported changes to the interventions alone. Even though researchers may themselves attribute these changes to the intervention, there may be a complex interplay of factors, e.g. the growth that happens during the work done for a PhD, contributing to their changing views and skill sets.

Authors’ response: We understand and see your point for this part to be less “causal”. To dampen our conclusion statement, we modified the conclusion as follows: “Being trained and coached in the conduct of preclinical SRs appears to be a contributing factor to many beneficial changes which will impact the quality of preclinical research in the long-term. Our findings suggest that this ZonMw funding scheme is helpful in improving the quality and transparency of preclinical research.”

4. General: It would be useful if authors could explicitly define early on in the manuscript what is meant by ‘preclinical’, as the term is also sometimes used to refer to pre-hospital and emergency medicine contexts.

Authors’ response: To clarify what we meant, we added to line 56-57: “SRs are struggling to achieve similar status in the preclinical field (i.e., fundamental and applied animal studies, in vitro and ex vivo studies before clinical research)”

5. Line 41: Suggest changing to ‘Keeping up to date with health/medical literature…’

Authors’ response: Agreed, the sentence was modified as suggested. 

6. Line 46: Please remove errant ‘-‘ from the sentence ending on this line.

Authors’ response: Thank you for noticing this mistake. It was deleted accordingly. 

7. Line 56: Consider changing ‘throw light’ to ‘cast light’.

Authors’ response: this sentence fragment was modified as suggested. 

8. Line 74: This introductory line should be presented as such. In its present state it is confusing and appears as another question, instead of the overarching aim. I would suggest amending this sentence to ‘This work aims to assess the impact of conducting preclinical SRs on researchers, their research and their field through the following objectives:’

Authors’ response: Indeed, the sentence suggested provides a clearer introductory line. We modified it as suggested, with a small modification (we changed “This work” to “Our study”). 

9. Lines 115-116: While it is acknowledged that these would be outside the scope of the current manuscript, which already describes a great deal of work, it would be interesting to assess the policy impacts and societal impacts (areas 2 and 4 in the manuscript) of the intervention in question using the modified Kuruvilla et al. 2007 checklist.

Authors’ response: We agree that policy impacts and societal impacts would have been very interesting to assess. However, our entire set of questions in both survey and interviews are based to match the research impact category of the Kuruvilla et al. 2007 checklist and the behavioural wheel of change. Assessing policy and societal impacts would require other questions and hence to perform a whole new study – which we cannot do in the scope of this manuscript. We do recognise the relevance of the other areas and hence have added a sentence to the discussion in that regard. The sentence is: “Further studies are warranted to fully understand what hinders or facilitate the conduct of preclinical SRs. For instance, a more in-depth assessment of impacts would provide greater insight and could address other areas of the research impact framework, e.g., policy impacts, societal impacts”.

10. Line 142: It is not clear what is meant by ‘thrive’ in this context, but it is assumed to be an incorrect translation. Consider changing to ‘drive’ or ‘development’, depending on the original intent.

Authors’ response: As the reviewer mentioned, it is an incorrect translation. After review, we modified thrive to ‘motivation’. 

11. Lines 157-158: This sentence contradicts methodology in following sections, which describes participants who had not finished their SRs also included in the sample. Please correct this factual inaccuracy, or elaborate on why non-completers were included post hoc – if that was the case.

Authors’ response: Participants were recruited regardless if they completed, started (and are currently conducting) or started and stopped their systematic review. We can see how the second part can be confusing/unclear in line 157-158. Therefore, we modified it as follows: “The target population for the questionnaires comprised researchers who had participated in a ZonMw workshop and who had either started (currently conducting or prematurely stopped) or completed a preclinical SR with funded coaching.”

12. Line 163: It may be useful to include here whether participants consented to be contacted by ZonMw for the purposes of intervention assessment when providing details, if this was indeed the case.

Authors’ response: ZonMw was in contact with all researchers by being their funder. By default, researchers agreed to communicate with ZonMw on their work and progress. Besides, they were given the opportunity to refuse participation or withdraw at any time. We are not sure if this information is required in the manuscript, as “at worst”, researchers received 2 e-mails kindly inviting them to participate in a survey – without any pressure-, which is not burdensome or intimate.

13. Line 177: It would be useful, though not essential, to state at the start of this section whether informed consent was sought to use the responses provided in questionnaires.

Authors’ response: The introduction text of the questionnaires clearly stated the intentions of the questionnaire. Though it was not exactly stated as such, we have assumed that researchers agree for their data (anonymously) to be collected, otherwise they would not have agreed to fill out the questionnaire in the first place. 

14. Line 186: As it is not possible for researchers still conducting their SRs to evaluate post-SR skills and experience, it is assumed that this should read ‘For the latter, only points 1, 2 and 3 were evaluated.’ Please correct if this is the case, or elaborate on how post-intervention experiences would be appraised before the end of the intervention – if the manuscript is correct in its current format.

Authors’ response: We realise that this part does not convey the information we wanted. We meant “skills gained as a result of conducting (steps of) the SR”. Considering that systematic reviews follow structured steps, one researcher could learn new skills by doing some of the steps without actually completing the whole study, e.g., designing a comprehensive search. It is indeed points 3,4, and 5 for the “ongoing group”. We added the following:

“4) skills gained as a result of conducting (steps of) the SR, and 5) experience with conducting (steps of) the SR (including, but not limited to, publishing experiences and wishing to perform further SRs for the completed group))”. 

15. IMPORTANT Lines 205-208: It is a pity that more demographic/explanatory variables were not collected on participants, as some measures of association could have been calculated. This would have provided some insights into differences between researchers and how this may, or may not, have shaped their responses – specifically given the outstanding uncertainty presented by the lack of a control group of questionnaire participants. It would be useful if the authors could include these considerations in their manuscript, perhaps as a next step in the evolution of understanding the role of SRs in preclinical work.

Authors’ response: We understand your point of view and added this point to the discussion

“ In this case study, we focussed primarily on our intervention, and it should be noted that the improvements observed cannot be only attributed to preclinical SRs as complex factors are also involved and may have influenced participants’ answers (e.g., personal growth and capacity, background, seniority). Further assessments may help to highlight or understand the intricated factors contributing to the beneficial effects.”

“in the future, when preclinical SRs become more established, it would be interesting to repeat this study on a broader scale […]. A wider population would also allow analysis by demographic group, e.g., the comparison of senior vs younger researchers, comparison between research field.”

16. Lines 235-236: Please provide a breakdown of the number of inputs per background/career level, i.e. ‘…Thorough piloting was performed for both questionnaires and semi-structured interviews by seven researchers knowledgeable in SRs with varying backgrounds and career levels, namely research assistant (n=x), PhD student (n=y) and professor (n=z).’ I don’t think it’s necessary to identify a person at a professorial level as ‘associate’ (or not).

Authors’ response: We modified this statement accordingly: “Thorough piloting was performed for both questionnaires and semi-structured interviews by seven researchers knowledgeable about SRs and from a variety of backgrounds and career levels, background and career levels, namely research assistants (n=2), PhD student (n=1), (associate), and professors (n=4).”

17. Lines 243-244: It is very possible that response and completion rates may be defined differently by different sources. A leading survey software defines response rate as ‘the number of people who completed the survey divided by the total sample’ (i.e. 45/95=47.4%), and completion rate as ‘the number of surveys filled out and submitted divided by the number of surveys started’ (i.e. 45/61=73.8%). Please revisit these concepts and calculations, and provide a reference for the definitions used to avoid confusion among readers.

Authors’ response: For clarity purposes, we decided to use the same definition you provided with a clear reference. Also, figure 2 was modified accordingly.

18. Line 248: Please revise the start of the sentence ‘And the 7 remaining…’. It is recommended that this sentence be combined with the previous sentence.

Authors’ response: we combined the two sentences as suggested. It now reads: “Two participants did not complete their SRs due to time constraints, while the 7 remaining participants terminated the questionnaire before answering the question about the state of their review”.

To note: We reverified all numbers and identified that an error got introduced. It was not 7 but 5 participants who terminated the questionnaire (the 5 participants were already indicated in Fig 2). It was modified appropriately.

19. Line 265-266: I am not convinced that reporting an ‘average median’ is wise; it is also a meaningless value in terms of agreement without information on the statements asked and the direction of the question. I wouldn’t risk readers skimming over this without referring to the figures, where the median breakdown per question provides much richer information.

Authors’ response: We see your point and hence removed the sentences mentioning average median. To give you some context, our intention with the average median was to give an overall value for each part, while bringing new information that was not mentioned in the figure. We agree with the comment and wish people would directly look in detail at the figures.

20. Line 317: Steer clear of statements such as ‘the results are quite positive’ as this indicates inherent bias for a result.

Authors’ response: We deleted this part and read the result thoroughly to remove similar statements. 

21. IMPORTANT Lines 317-321: As stated before, it is very difficult to know whether the equivalence in the response of SR completers and non-completers can be ascribed to the SR, given the lack of a control comparator. It is possible that these insights may have evolved in respondents in a similar field, and with comparable education and training, in the absence of conducting a preclinical SR. This evolution might even be suggested by the identical medians in the two groups, with further SR experience (in completers) leading to no increase in critical appraisal compared to non-completers. It is accepted that this uncertainty is part of the case study design, but please make this clear to the reader.

Authors’ response: We added a sentence about this matter in the discussion: “Moreover, our study design does not include a control group nor a pre-post intervention assessment, which limits the certainty one can have on the causality of our intervention. An example of this uncertainty can be observed in Fig 4, where both groups answered similarly regardless of whether they completed their review. This highlights again that the impacts and benefits found in our study cannot be only attributed to the conduct of preclinical SR but may be influenced by the workshop. We could imagine that similar education, i.e., workshops with hands-on practice, could produce similar results without actually conducting a preclinical SR.”

“Taken together, the intervention seems promising but our findings need to be interpreted within the narrow scope of the case study”

22. IMPORTANT Line 338: It would be interesting to know whether respondents participating in the scheme during 2020 had a different experience to those pre-pandemic in terms of time taken/delays with their SRs. If these data are available this might be an interesting aspect to include, as the time required to conduct an SR often presents a considerable barrier to their initiation and completion.

Authors’ response: It would be an interesting comparison to make, however we do not have the data to perform this analysis completely. Nevertheless, three participants (one from the completed group and two from the started group) mentioned delays in relation to the covid situation. To address delays, we added the following paragraph “Reviews (and reviews steps) often took longer than expected (26/30 participants for the completed group and 8/15 participants for the ongoing group). For instance, researchers often experienced the number of studies during screening, data extraction or data analysis as overwhelming, but also suffered in getting their protocols, manuscripts or revision for publication reviewed. Some had to wait for second or third assessors/reviewers and experienced delays when asking authors for further information. Also, three participants (both groups combined) mentioned the COVID-19 pandemic as a delaying factor.”

23. Lines 470 and 484: Please steer clear from informal language such as ‘no easy business’ and ‘budge one bit’. Furthermore, ‘(yet)’ [line 484] again demonstrates an inherent bias by the author for a certain outcome, and should be avoided at all costs.

Authors’ response: These instances were removed and modified as follow: “Interviewees’ experiences highlighted that getting preclinical SRs published can be challenging.” We also added: “discovering that the insights from their SRs appeared to have no impact on their field.” 

We verified the rest of the manuscript text to remove informal language. 

Regarding the ‘(yet)’, we meant that the participants wished for changes in their field (at least by the ones that were frustrated) but that this change did not yet occur. However, we see how it could be misleading and thus removed it according to your suggestion. 

24. IMPORTANT General: As stated before, it is very difficult to know whether the reported experiences and skills of respondents can be ascribed to conducting the preclinical SR, given the lack of a control comparator or association with explanatory variables. It is possible that the reported insights would have developed in respondents in a similar field and with comparable education and training. The authors do address the subjective nature of their findings, but could suggest next steps in this important work. As a starting point, it might be useful to suggest (and consider) a pre-post test with a new intake of researchers, measuring their attitudes and skills at baseline and again following the completion of their SR. An alternative, but a weaker test of causality, would be to measure explanatory variables and test associations with reported outcomes in future work.

Authors’ response: We added several statements to clarify this aspect: 

-line 596-600: In this case study, we focussed primarily on our intervention, and it should be noted that the improvements observed cannot be only attributed to preclinical SRs, as complex factors are also involved (e.g., personal growth and capacity, background, seniority). Further assessments may help to highlight or understand the complex factors contributing to the beneficial effects.

-line 647-648: In addition, a pre-post test study could be performed to evaluate changes in skills and behaviour before and after the intervention.

25. General: It was surprising not to see any exploration of the effect of the pandemic on the findings presented in this paper, given both the global impact of the event as well as several mentions of COVID-19 in the questionnaire responses – specifically related to delays in SR completion. It might be useful for the authors to include a paragraph on how this context may have shaped their findings, both in terms of practicalities for SRs as well as responses; particularly due to shifts in mental health and outlook for many during this time.

Authors’ response: It is indeed an important factor, we now mention it in the strengths and limitations, with the following paragraph: “Second, our data collection took place during the COVID-19 pandemic and the study design we used limits the generalisation of our findings. Our findings are reported impacts, thus they are subjective to the perceptions of the participants, in contrast to measurable, tangible impacts. As mentioned in the result, the fact that (part of the) participants conducted their review during the COVID-19 pandemic may have impacted their experience and hence influenced their answers. On the one hand, researchers might have had more time or opportunities to work on meta-research during the pandemic. On the other hand, the pandemic conditions seem to have created delays in conducting the SRs and could have impacted the researchers personally, including their capacity and opportunity for professional development.” 

26. IMPORTANT Lines 568-569: Statements attributing changes in mindset and behaviour to the conduct of preclinical SRs without acknowledging the complex myriad of circumstances that may contribute to these are problematic. Please review this statement (and others like it) to accurately reflect that preclinical SRs *appear to contribute* to a shift in mindset and behaviour, though comparative data, and more work to identify other explanatory factors, is needed.

Authors’ response: We overall reduced the strength of the language when talking about causality, for example saying “contribute to” or “seem to participate” instead of “trigger” or “enable”. In addition, as mentioned in an earlier comment, we added the following sentence: “it should be noted that the improvements observed cannot be only attributed to preclinical SRs, as complex factors are also involved (e.g., personal growth and capacity, background, seniority). Further assessments may help to highlight or understand the complex factors contributing to the beneficial effects.”

27. Conflicts of interest. The authors declare their association with the invention provider upfront, and no serious conflicts were identified. I would like to caution the authors, however, around the language they use and how it unintentionally reflects a preferred direction of their findings.

Authors’ response: We performed a thorough check of the manuscript to ensure that no preferred directions were mentioned in the findings. When necessary, we rephrased/removed part of the text. Of course, positive language was not consciously included in the text. We acknowledge our enthusiasm for systematic reviews, which may have unintentionally been included in our writing process. We would like to thank you for your warning and have rewritten the text accordingly.

To note: We ran checks for minor spelling, punctuation and grammatical errors as you suggested.

---

## [Decision Letter · Decision Letter 1]

3 Nov 2021

PONE-D-21-16684R1The impact of conducting preclinical systematic reviews on researchers and their research: a mixed method case studyPLOS ONE

Dear Julia Menon,

Thank you for submitting your manuscript to PLOS ONE.  Your responses to the previous comments raised have been well received. We however request that you respond to two minor comments by reviewer 2. Therefore, we invite you to submit a revised version of the manuscript that addresses the points raised during the review process.

 Please submit your revised manuscript by 18 Dec 2021. If you will need more time than this to complete your revisions, please reply to this message or contact the journal office at plosone@plos.org. Please include the following items when submitting your revised manuscript:A rebuttal letter that responds to each point raised by the academic editor and reviewer(s). You should upload this letter as a separate file labeled 'Response to Reviewers'.A marked-up copy of your manuscript that highlights changes made to the original version. You should upload this as a separate file labeled 'Revised Manuscript with Track Changes'.An unmarked version of your revised paper without tracked changes. You should upload this as a separate file labeled 'Manuscript'.We look forward to receiving your revised manuscript.

Kind regards,

Eleanor Ochodo

Academic Editor

PLOS ONE

Journal Requirements:

Reviewers' comments:

Reviewer's Responses to Questions

**Comments to the Author**

1. If the authors have adequately addressed your comments raised in a previous round of review and you feel that this manuscript is now acceptable for publication, you may indicate that here to bypass the “Comments to the Author” section, enter your conflict of interest statement in the “Confidential to Editor” section, and submit your "Accept" recommendation.

Reviewer #1: All comments have been addressed

Reviewer #2: (No Response)

2. Is the manuscript technically sound, and do the data support the conclusions?

Reviewer #1: Yes

Reviewer #2: Yes

3. Has the statistical analysis been performed appropriately and rigorously? 

Reviewer #1: Yes

Reviewer #2: Yes

4. Have the authors made all data underlying the findings in their manuscript fully available?

Reviewer #1: Yes

Reviewer #2: Yes

5. Is the manuscript presented in an intelligible fashion and written in standard English?

Reviewer #1: Yes

Reviewer #2: Yes

6. Review Comments to the Author

Reviewer #1: (No Response)

Reviewer #2: Thank you very much for your considered feedback and amendments to the manuscript - as I stated during the first round of review, this is a very important piece of work; I hope to see it in print (and cite it!).

I have no further *material* comments, but two further suggestions to consider, based on your feedback:

- Line 249: I think '(associate)' can be removed from the text entirely, i.e. "...namely research assistants (n=2), a PhD student (n=1) and professors (n=4)."

- Line 523: I take your point on the intention behind "...had not (yet) made..." in the previous version, and that this reflects the wishes of the participant rather than you as an author team. While I do think it's better to steer clear from any wording that could result in perceived bias, you could consider changing the sentence to "...discovering that the insights from their SRs appeared to have had no impact on their field to date." This is entirely up to you, as the revised version you provided is also fine, but perhaps this provides a bit more nuance.

Wishing you all the best,

Amanda

7. PLOS authors have the option to publish the peer review history of their article (what does this mean?). If published, this will include your full peer review and any attached files.

Reviewer #1: No

Reviewer #2: **Yes: **Dr Amanda Salomé Brand

---

## [Author Response · Author response to Decision Letter 1]

11 Nov 2021

Reviewer’s comments

1. Line 249: I think '(associate)' can be removed from the text entirely, i.e. "...namely research assistants (n=2), a PhD student (n=1) and professors (n=4)."

Authors’ response: We agree that this term makes the sentence “wordy” and does not per se add to the context. We have removed it as suggested. 

2. Line 523: I take your point on the intention behind "...had not (yet) made..." in the previous version, and that this reflects the wishes of the participant rather than you as an author team. While I do think it's better to steer clear from any wording that could result in perceived bias, you could consider changing the sentence to "...discovering that the insights from their SRs appeared to have had no impact on their field to date." This is entirely up to you, as the revised version you provided is also fine, but perhaps this provides a bit more nuance.

Authors’ response: Thank you for this suggestion! We feel it indeed brings more nuance and clarity to this part of the text. The sentence was modified as suggested.

---

## [Editor Report · Decision Letter 2]

15 Nov 2021

The impact of conducting preclinical systematic reviews on researchers and their research: a mixed method case study

PONE-D-21-16684R2

Dear Julia Menon,

We’re pleased to inform you that your manuscript has been judged scientifically suitable for publication and will be formally accepted for publication once it meets all outstanding technical requirements.

Kind regards,

Eleanor Ochodo

Academic Editor

PLOS ONE

---

## [Editor Report · Acceptance letter]

1 Dec 2021

PONE-D-21-16684R2 

The impact of conducting preclinical systematic reviews on researchers and their research: a mixed method case study 

Dear Dr. Menon:

I'm pleased to inform you that your manuscript has been deemed suitable for publication in PLOS ONE. Congratulations! Your manuscript is now with our production department. 

Kind regards, 

on behalf of

Prof Eleanor Ochodo 

Academic Editor

PLOS ONE